# Sex- and Tissue-Specific Effects of Leukemia Inhibitory Factor on Mitochondrial Bioenergetics Following Ischemic Stroke

**DOI:** 10.3390/biom15050738

**Published:** 2025-05-20

**Authors:** Hemendra J. Vekaria, Sarah J. Shelley, Sarah J. Messmer, Prashant D. Kunjadia, Christopher J. McLouth, Patrick G. Sullivan, Justin F. Fraser, Keith R. Pennypacker, Chirayu D. Pandya

**Affiliations:** 1Spinal Cord and Brain Injury Research Center (SCoBIRC), University of Kentucky, Lexington, KY 405036, USA; vekaria@musc.edu (H.J.V.); patsullivan@uky.edu (P.G.S.); 2Lexington Veterans Affairs Healthcare System, Lexington, KY 40502, USA; 3Department of Neurosurgery, Medical University of South Carolina, 173 Ashley Ave, Charleston, SC 29425, USA; 4Department of Neurosurgery, University of Kentucky, Lexington, KY 40506, USA; sjsh254@uky.edu (S.J.S.); sarah.goodwin2@uky.edu (S.J.M.); pdku223@uky.edu (P.D.K.); jfr235@uky.edu (J.F.F.); 5The Center for Advanced Translational Stroke Science, University of Kentucky (CATSS), Lexington, KY 40506, USA; 6Department of Neuroscience, University of Kentucky, Lexington, KY 40506, USA; 7Department of Neurology, University of Kentucky, Lexington, KY 40506, USA; cmclouth@uky.edu; 8Department of Biostatistics, University of Kentucky, Lexington, KY 40506, USA; 9Department of Radiology, University of Kentucky, Lexington, KY 40506, USA; 10Department of Otolaryngology, University of Kentucky, Lexington, KY 40506, USA

**Keywords:** stroke, MCAO, mitochondrial bioenergetics, striatum, prefrontal cortex, LIF

## Abstract

Oxidative stress due to increased reactive oxygen species (ROS) formation and/or inflammation is considered to play an important role in ischemic stroke injury. Leukemia inhibitory factor (LIF) has been shown to protect both oligodendrocytes and neurons from ischemia by upregulating endogenous anti-oxidants, though the effect of ischemia and the protective role of LIF treatment in mitochondrial function have not been studied. The goal of this study was to determine whether LIF protects ischemia-induced altered mitochondrial bioenergetics in reproductively senescent aged rats of both sexes (≥18 months old), approximately equivalent to the average age of human stroke patients. Animals were euthanized at 3 days after permanent middle cerebral artery occlusion (MCAO) surgery. We found that MCAO surgery significantly reduced mitochondrial oxidative phosphorylation in both the ipsilateral striatum and prefrontal cortex in male aged rats compared to their respective contralateral regions of the brain. MCAO injury showed mitochondrial bioenergetic dysfunction only in the striatum in female rats; however, the prefrontal cortex remained unaffected to the injury. LIF-treated rats significantly prevented mitochondrial dysfunction in the striatum in male rats compared to their vehicle-treated counterparts. Collectively, MCAO-induced mitochondrial dysfunction and LIF’s potential as a therapeutic biomolecule exhibited sex- and tissue-specific effects, varying between the striatum and prefrontal cortex in male and female rats.

## 1. Introduction

Stroke remains a leading cause of death and disability worldwide and can be categorized into two main subtypes. Roughly 70% of stroke occurrences are those of the ischemic subtype, which occurs when a cerebral blood vessel undergoes thrombosis. Although many strategies such as neuroprotection, neurorepair, and stem cell therapy have been explored, intravenous thrombolysis (IVT) and mechanical thrombectomy (MT) are the only standard treatment therapies for stroke [1,2]. Both treatments must be administered within a limited time after the onset of a stroke and must meet specific imaging criteria [3,4]; otherwise, patients receive only supportive medical care [5]. The proportion of stroke patients receiving IVT and/or MT is small because of issues related to access, exclusion criteria, and time windows; therefore, new and more accessible treatments are critically needed.

There are different mechanisms involved in the pathogenesis of stroke, and increasing evidence shows that energy imbalance and reactive oxygen species (ROS) significantly account for the progression of ischemic stroke. Oxidative stress, inflammation, and energy failure due to mitochondrial dysfunction are significantly associated with the ischemic condition, which leads to membrane depolarization [6], calcium influx, and the activation of pro-oxidant enzymes [7,8]. Mitochondria are the most critical regulators of cell survival and death due to their roles in the maintenance of cellular energy homeostasis and regulation of redox-dependent signaling pathways [9]. Moreover, mitochondrial abnormalities are frequently observed in demyelinated axons and neuronal cell bodies [10]. Mitochondrial dysfunction is attributed to the excitotoxicity of neurons triggered after the injury, where the activation of NMDA receptors leads to increased Ca^2+^ influx. Excess Ca^2+^ is sequestered by mitochondria. Due to the ROS and inflammatory response the calcium carrying capacity of the mitochondria gets decreased, leading to the premature activation of the mitochondrial permeability transition pore (mPTP) leading to cell death signaling [11]. Our previous studies demonstrated that TNFα overexpression leads to increased mitochondrial dysfunction and impaired calcium buffering during ischemia [12]. These findings suggest that an elevated inflammatory response following neuronal injury exacerbates damage through mitochondrial dysfunction, further amplifying the extent of ischemic injury. Dysfunctional mitochondria can also lead to the increased presence of ROS that induce the peroxidation of lipids, proteins, and DNA, resulting in axonal energy failure and subsequent neurodegeneration [13]. However, the mechanisms underlying the interactions between oxidative stress, inflammatory response, and brain mitochondrial bioenergetic pathways, and their relevance to ischemic stroke are not fully characterized.

Leukemia inhibitory factor (LIF) has shown promising results as a therapeutic drug in several animal models of neuronal injury [14,15,16,17]. LIF is an anti-inflammatory interleukin-6-class cytokine and plays an important role in cell differentiation and growth. LIF binds to the specific LIF receptor (LIFR-α), which forms a heterodimer with a specific subunit common to all members of that family of receptors, the GP130 signal transducing subunit [18]. The binding of LIF to its receptor leads to the activation of the JAK/STAT (Janus kinase/signal transducer and activator of transcription) and MAPK (mitogen-activated protein kinase) downstream signaling pathways [15]. Circulating LIF crosses the blood–brain barrier (BBB) via a saturable transport mechanism, enabling central nervous system responses to various stimuli [14]. Previous reports have shown that the administration of LIF reduces demyelination by enhancing oligodendrocyte survival in the experimental model of multiple sclerosis (MS) [15] and a murine model of spinal cord injury (SCI) [16]. In addition, LIF reduced the degeneration of motor neurons in the SOD1 G93A murine model of familial amyotrophic lateral sclerosis (ALS) [19]. Although LIF therapy showed beneficial effects as a treatment for several neurological disorders, its association with pathophysiology in ischemic stroke is poorly understood. Our group first showed that LIF decreases the infarct volume, prevents oligodendrocyte death, and improves functional recovery in rat models of focal ischemia [20]. Recently, we showed LIF protects neurons in vivo and in vitro from ischemic injury via the upregulation of anti-oxidant enzyme SOD3 [21]. However, the role of LIF in targeting mitochondrial dysfunction to impart neuroprotection from ischemic stroke injury has not yet been studied.

Recognizing the critical role of mitochondrial dysfunction in assessing neuroprotective therapies after ischemic stroke, it is instrumental to find out how LIF alters the metabolic regulation in a tissue-specific manner. To test this, we designed this study to provide a detailed sex- and region-specific evaluation of LIF’s therapeutic potential on mitochondrial dysfunction following middle cerebral artery occlusion (MCAO) in aged rats.

## 2. Materials and Methods

### 2.1. Experimental Overview

We assessed mitochondrial bioenergetics parameters—State III (ATP synthesis rates), State IV (basal respiratory capacity), and State V (maximal respiratory capacity)—in the isolated mitochondria from striatum and prefrontal cortex in both male and female aged rats using the Seahorse Flux Analyzer (Agilent Technologies, Santa Clara, CA, USA). Additionally, we explored the potential therapeutic effects of LIF on mitochondrial bioenergetics following ischemic stroke. The motor and cognitive function were also evaluated before and after stroke induction in both phosphate-buffered saline (PBS)-, served as vehicle, and LIF-treated groups. These findings contribute to a stronger foundation for evaluating mitochondrial-targeted therapies in experimental ischemic stroke models. A conceptual diagram illustrating the design of the present study is shown in Figure 1.

### 2.2. Animal Model Selection and Care

Stroke subtype, complex cerebrovascular anatomy, and comorbidities associated with aging can make modeling human stroke particularly difficult. These extraneous variables may significantly alter cellular and molecular responses to ischemic injury. To model ischemic stroke in the most commonly affected human population (50–60 years old), in this study, we employed ~18-month-old aged Sprague-Dawley rats (*n* = 19; 10 males, 9 females, Envigo, Indianapolis, IN, USA). The minimum quantity of animals required for each study was determined a priori using a power analysis as per our previously published report [17]. Animals were approved by the Institutional Animal Care and Use Committee of the University of Kentucky. Animals were provided with constant access to food and water, kept in a climate-controlled room, and maintained on a 12 h light/dark schedule. Careful planning and study design mitigated pain, distress, and discomfort.

### 2.3. Surgery Preparation

Prior to surgery, rats were placed in an induction chamber using a scavenger-equipped calibrated vaporizer and anesthetized using 5% isoflurane with 1 L per minute (LPM) of oxygen. Once fully anesthetized, each animal was removed from the chamber and placed on a rodent-specific nose cone, and the dose of isoflurane was lowered to 2–3% with a flow rate of 1 L of oxygen per minute for the duration of surgery. The superior and anterior cervical region of the head and neck were shaved, cleaned with Hibiclens (chlorhexidine scrub) three times, and rinsed with 70% EtOH, and we applied a betadine solution. Subcutaneously administered Carprofen (Rimadyl, 50 mg/mL diluted with saline to 5 mg/mL; dosed at 5 mg/kg s.c.) and Atropine (0.54 mg/mL diluted with saline to 0.108 mg/mL; dosed at 0.04 mg/kg s.c.) provided pre-operative analgesia. Surgical tape secured the forelimbs and tail to an elevated platform for the surgery.

### 2.4. Middle Cerebral Artery Occlusion

Permanent MCAO procedures induced permanent focal ischemia as previously described [16]. Briefly, a 40 mm monofilament was secured approximately 20 mm from the base of the MCA, via introduction first into the ECA, then distally into the ICA. Prior to the advancement of the monofilament into the MCA, a micro-serrefine arterial clamp (FST, Fine Science Tools #18,055-01) occluded the internal monitored carotid artery (ICA) and common carotid artery (CCA). One 18-inch length of 5-0 silk suture ligated the external carotid artery (ECA), and another equal-length suture secured the monofilament within the ICA. Incisions were closed with 3-0 nylon monofilament and rats were placed in clean recovery cages.

### 2.5. Post-Operative Animal Care and LIF Treatment

Rats were closely during surgical recovery and were provided subcutaneous saline injections to mitigate dehydration. Once the animals were fully awake and sternal, they were returned to their housing room. Critical post-operative care protocol included pain management and recovery assistance continued for 72 h. Animals were intravenously administered either PBS (pH 7.4) or 125 μg/kg LIF at 6, 24, and 48 h after surgery.

### 2.6. Chemical Preparation

#### 2.6.1. Mitochondrial Substrates and Modulators

Substrate stocks prepared for mitochondrial bioenergetics assessment included 500 mM pyruvate, 250 mM malate, 150 mM ADP, 10 mg/mL oligomycin A, 10 mM carbonyl cyanide 4-(trifluoromethoxy) phenylhydrazone (FCCP), and 1 M succinate. The aqueous stocks were adjusted to pH 7.2 and all the stocks were stored at −20 °C.

#### 2.6.2. Mitochondrial Isolation Buffer (MIB)

Throughout the isolation procedure, materials were maintained at 4 °C. MIB (pH 7.2) was composed of 215 mM of mannitol, 75 mM of sucrose, 0.1% bovine serum albumin (BSA), 1 mM ethylene glycol-bis (β-aminoethyl ether)-N,N,N′,N′-tetraacetic acid (EGTA), and 20 mM N-2-hydroxyethylpiperazine-N′-2-ethanesulfonic acid (HEPES).

#### 2.6.3. Mitochondrial Respiration Buffer (RB)

The RB composed of 125 mM KCl, 0.1% BSA, 20 mM HEPES, 2 mM MgCl_2_, and 2.5 mM KH_2_PO_4_, adjusted to pH 7.2 with KOH, was stored at 4 °C. All mitochondrial working stocks were prepared in RB and the amount of each substrate used in respiration measurements listed in subsequent sections is based on the 175 μL RB volume.

### 2.7. Mitochondrial Isolation

To isolate mitochondria from rat brain tissue samples, the rats were euthanized using CO_2_ and the brain was quickly removed from euthanized animals and placed in a stainless-steel brain matrix. The prefrontal cortex and striatum brain tissues were dissected from both contralateral and ipsilateral hemispheres and immediately submerged in 2 mL of cold mitochondria isolation buffer (pH 7.2) and homogenized. Homogenized tissues were then centrifuged at 1300 rcf for 3 min at 4 °C, and the supernatant was collected into new 2 mL tubes. The new tubes were centrifuged at 13,000 rcf for 10 min at 4 °C. The supernatants were discarded, leaving the pellets that were resuspended in 450 μL MIB and were subjected to pressurized nitrogen cell cavitation at 1000–1200 psi (Parr instruments #4635) for 10 min to release synaptic mitochondria. Following the nitrogen cavitation, homogenates were transferred to 1.5 mL tubes topped off with MIB and final centrifugation was performed at 10,000 rcf for 10 min at 4 °C. The supernatants were discarded and the pellets were resuspended in 50–100 μL MIB for mitochondrial protein estimation using BCA method (Pierce™ BCA Protein Assay Kits #23227) to optimize the loading for bioenergetic assessments.

### 2.8. Mitochondrial Respiration Measurements

To measure respiration rates, mitochondria were isolated from the striatum and the prefrontal cortex regions of both contralateral and ipsilateral sides of the rat brains from MCAO rats treated with PBS or LIF. The bioenergetic assessments were carried out using XFe96 Seahorse Extracellular Flux Analyzer (Agilent Technologies). Our selection of state parameters represented mitochondrial bioenergetic function during the oxidative phosphorylation. State III was measured by adding excess ADP in the presence of pyruvate and malate, stimulating mitochondria to drive ATP production capacity through complex I. Once this excess ADP was consumed or depleted from State III, oligomycin-mediated inhibition of FoF1 ATPase led to State IV measurements deriving the leaky respiration, which was helpful in identifying the extent of leak in the mitochondrial membrane permeability. State V-CI was measured by adding the uncoupler, FCCP (Carbonyl cyanide 4-(trifluoromethoxy)phenylhydrazone), yielding the maximal respiration capacity mediated by complex I (CI). Lastly, addition of rotenone shut down complex I and succinate-driven State V-CII assessed mitochondrial respiration mediated by FADH_2_ instead of NADH, which provided insight into maximal respiration capacity supported by complex II (CII).

The respiration measurements were carried out as previously described [22,23,24,25,26] with some modifications. Briefly, one day prior to mitochondrial isolation, a XFe96 flux assay plate was hydrated and stored in a non-CO_2_ incubator at 37 °C. On the day of analysis, sensor cartridge ports were loaded with 25 μL each of the corresponding experimental reagents to encounter the final concentrations in the parenthesis—port A: a mixture of pyruvate (5 mM), malate (2.5 mM), and ADP (2 mM); port B: oligomycin A (2.5 μM); port C: FCCP (4 μM); and port D: rotenone (0.8 μM) and succinate (10 mM). Once loaded, the assay plate was placed into the Seahorse XFe96 Flux Analyzer for automated calibration.

Isolated mitochondria from ipsilateral and contralateral striatum and prefrontal cortex brain regions (8 μg/well) were loaded onto a XF96 cell culture plate and were analyzed for the oxygen consumption rates. One plate each was run for the prefrontal cortex and striatum. The respiration rates were measured as State III (complex-I-driven respiration using PM + ADP), State IV (leak respiration after oligomycin injection), State V-CI (maximal complex-I-driven uncoupled respiration after FCCP injection), and State V-CII (maximal complex-II-driven uncoupled respiration using rotenone and succinate).

### 2.9. Neurological Assessment Score

Our selection of behavioral tests, as previously described [20], aimed to identify functional deficits associated with motor function. Baseline assessments were performed prior to surgery for all tests except circling, and post-surgical assessments immediately preceding euthanasia, approximately 72 h after ischemic model surgery. Circling tests were conducted by placing rats in an open field and allowing them to move freely. Circling behavior was recorded as absence (“No”) or presence (“Yes”) over a 5 min testing period. Baseline assessment was not performed for this test since normal, healthy rats do not exhibit this behavior. Paw extension tests were used to evaluate restricted forelimb mobility resulting from ischemic injury and is achieved by suspending the rat above an open cage over a 3 min testing period, observing its ability (“Yes”) or inability (“No”) to fully extend both forelimbs without impairment. An elevated body swing test (EBST) records the number of swings and direction a rat completes, attempting to correct itself, while being suspended by the tail over an empty cage. This test was conducted for 10 trials (about 10 s/trial) over a 5 min period. Lastly, step tests helped to evaluate motor symmetry by re-straining all but one forelimb at a time and pulling the animal in the direction of the unrestrained forelimb over a hard, flat surface. The number of steps were recorded over 1 m distance at a constant speed in both directions over a 10 min testing period. The circling and paw extension tests were expressed as frequency values, and EBST and step tests were calculated as absolute bias to demonstrate asymmetrical motor function.

### 2.10. Statistical Analysis

Oxygen consumption rates (OCRs) are presented as bar graphs showing individual data points and error bars represented by ± the standard error of the mean (SEM) of percentages of the OCR rate of PBS-treated contralateral regions. Mitochondrial samples were prepared as described previously from individual LIF- and PBS-treated male and female rats (*n* = 4–5 per group) were analyzed separately using contralateral and ipsilateral striatum and prefrontal cortex region tissues. Comparisons for OCR analysis were performed using a mixed-effects model, including between-subject effects (treatments) and within- subject effects (contralateral vs. ipsilateral) followed by a Tukey’s multiple comparison post hoc analysis with SAS version 9.4 (SAS Institute, Inc., Cary, NC, USA). One ipsilateral sample retrieved from a PBS-treated male was identified as an outlier and excluded from analysis. Circling and paw extension test results were evaluated using a chi-square analysis. Comparison for step and elevated body swing tests were performed using a two-way analysis of variance (ANOVA) followed by a Tukey’s multiple comparison post hoc analysis. (GraphPad Prism 9 software package; GraphPad Software Inc., La Jolla, CA, USA). Statistical significance was defined as *p* ≤ 0.05.

## 3. Results

### 3.1. LIF Restores Stroke-Induced Decrease in Mitochondrial Respiration in Striatum in Male Rats

To measure the respiration rates of electron transfer pathway states III, IV, V-CI, and V-CII, isolated mitochondria from homogenized brain tissues of contralateral (non-stroke) and ipsilateral (stroke) regions of the striatum were collected from male MCAO rats treated with PBS (*n* = 4–5) and LIF (*n* = 5) (Figure 2a–d). Apart from State IV, all mitochondrial oxygen consumption rates (OCRs) were lower in ipsilateral regions compared with contralateral regions in the male striatum in PBS-treated rats (Figure 2; State III: F_1,7_ = 49.62, *p* = 0.0004; State V-CI: F_1,7_ = 16.56, *p* = 0.0104; State V-CII: F_1,7_ = 39.57, *p* = 0.0005). No significant difference between contralateral and ipsilateral striatum OCRs was measured in LIF-treated male rats (Figure 2; *p* ≥ 0.2883). In the striatum state parameters III, IV, and V-CII, ipsilateral regions of LIF-treated males showed significantly higher OCRs than in PBS-treated males (State III: F_1,8_ = 16.07, *p* = 0.0030; State IV: F_1,8_ = 13.09, *p* = 0.0305; State V-CII: F_1,8_ = 12.74, *p* = 0.0027). Ipsilateral OCRs in LIF-treated rats approached significantly higher values than ipsilateral PBS-treated rats in State V-CI (F_1,8_ = 4.67, *p* = 0.0524).

### 3.2. Effect of LIF on Mitochondrial Respiration in the Prefrontal Cortex in Male Rats

In PBS-treated males, ipsilateral regions of the prefrontal cortex showed lower OCRs than contralateral regions in State IV (Figure 3b; F_1,7_ = 29.64, *p* = 0.0311). Ipsilateral prefrontal cortex regions in LIF-treated males showed lower OCRs than contralateral regions in State IV (Figure 3b; F_1,7_ = 29.64, *p* = 0.0202). States III, V-CI, and V-CII in LIF-treated males showed no significant difference between the contralateral and ipsilateral OCR of the prefrontal cortex (Figure 3; F_1,7_ = 10.11, *p* = 0.3958; F_1,7_ = 1.6, *p* = 0.9997; and F_1,7_ = 15.23, *p* = 0.1283, respectively). No significant difference was measured in any state parameter between PBS- and LIF-treated male ipsilateral prefrontal cortex OCRs (Figure 3; *p* ≥ 0.3360).

### 3.3. Effect of LIF on Mitochondrial Respiration in Striatum in Female Rats

In LIF-treated females (*n* = 5), the ipsilateral striatum showed a lower OCR than in contralateral regions in State V-CII (Figure 4d; F_1,7_ = 17.72, *p* = 0.0469) while States III and V-CI approached significantly lower OCRs (Figure 4; State III, F_1,7_ = 12.22, *p* = 0.0836; State V-CI, F_1,7_ = 13.36, *p* = 0.0771). No significant difference was measured in any state parameter between PBS- and LIF-treated female ipsilateral striatum OCRs (Figure 4; *p* ≥ 0.9704).

### 3.4. Effect of LIF on Mitochondrial Respiration in Prefrontal Cortex in Female Rats

Contralateral and ipsilateral regions of the PBS-treated female prefrontal cortex showed no significant difference in the OCR in any state parameters (Figure 5; *p* ≥ 0.5009). Ipsilateral prefrontal cortex regions approached significantly lower OCRs than in contralateral regions in LIF-treated females in State V-CII (Figure 5d; F_1,7_ = 7.99, *p* = 0.07162). In States III, IV, and V-CI, no significant difference was measured between the contralateral and ipsilateral regions of LIF-treated females (Figure 5; *p* ≥ 0.2036). No significant difference was measured in the ipsilateral prefrontal cortex OCR between PBS- and LIF-treated females for any state parameter (Figure 5; *p* ≥ 0.4061).

### 3.5. Effect of LIF Treatment on Neurological Function and Behavior

Functional deficits and recovery were evaluated using neurological motor assessment tests (described in Section 2.4) and compared with baseline and post-surgical test performances (Figure 6). For circling and paw extension tests, all baseline testing demonstrated normal behaviors were measured as normal (with “Yes” responses indicating the normal ability to extend the paw and “No” responses indicating the absence of abnormal circling behavior). Circling behavior did not significantly differ based on sex (χ^2^ = 2.574, df = 1, *p* = 0.1087) or treatment (χ^2^ = 2.121, df = 1, *p* = 0.1453) (Figure 6a). Post-surgical paw extension ability showed no significant difference based on sex (χ^2^ = 2.898, df = 1, *p* = 0.0887) or treatment (χ^2^ = 1.154, df = 1, *p* = 0.2827). Baseline step test outcomes did not significantly differ between sexes (F_1,32_ = 0.6105, *p* = 0.3882) or treatments (F_1,22_ = 0.2040, *p* = 0.9720). Differences in the post-surgical outcomes of the step tests did approach significance between sexes, with males exhibiting higher bias than females (F_1,32_ = 4.074, *p* = 0.0564), though no significant difference was detected between treatments (F_1,22_ = 0.2519, *p* = 0.5072). Further, the post-surgical step test bias significantly increased from baseline results in both males (F_1,22_ = 63.68, *p* < 0.0001) and females (F_1,22_ = 0.2519, *p* = 0.0464) and in both LIF- (F_1,32_ = 24.49, *p* < 0.0001) and PBS-treated rats (F_1,32_ = 24.49, *p* < 0.0001) (Figure 6c). Elevated body swing test (EBST) results showed no significant difference in bias between baseline and post-surgery results in males (F_1,32_ = 0.5633, *p* = 0.6858), females (F_1,32_ = 0.5633, *p* = 0.5183), and LIF-treated (F_1,24_ = 0.3666, *p* = 0.7579) or PBS-treated (F_1,24_ = 0.3666, *p* = 0.6059) rats. No significant difference was detected in EBST bias in post-surgical outcomes based on sex (*p* = 0.5716) or treatment (*p* = 0.3205) (Figure 6d).

## 4. Discussion

In this study, we investigated the potential neuroprotective role of leukemia inhibitory factor (LIF) via improving the mitochondrial respiration and neurological outcomes following ischemic injury. Previously, LIF, a neuroprotective and anti-inflammatory cytokine, was shown to decrease neurodegeneration and increase survival after a MCAO stroke model surgery in rats [20]. Our results align with these findings and provide further evidence that LIF treatment can positively influence in vivo mitochondrial respiration in MCAO stroke models, suggesting LIF’s role in protecting against ischemia-induced mitochondrial function damage.

A key area of focus in this study examined the effect of LIF on mitochondrial function in striatum and prefrontal cortex brain regions affected by ischemia in aged male and female rat models. After MCAO surgery, ipsilateral striatum tissues confirmed an injury effect, measuring significantly lower mitochondrial respiration compared to the contralateral striatum in both male and female rats in all states of respiratory parameters. Our previous sex-specific studies in the controlled cortical impact (CCI) model of traumatic brain injury (TBI) established that there are subtle differences in the synaptic and non-synaptic mitochondrial function after CCI in a time-dependent manner [23]. Our results with the current study suggest that MCAO-induced mitochondrial dysfunction responses also were sex- and tissue-specific, evident differentially at the striatum and prefrontal cortex in male and female aged rats. The current results strengthen the hypothesis and provide the basis that post-stroke mitochondrial dysfunction may be targeted differentially based on tissue specificity. Although the mitochondrial deficits in response to the neuronal injury constitute a well-known fact, the study of the mitochondrial flux still remains one of the most important indicators of early secondary injury. The drugs that directly or indirectly target these fluxes have proven to have positive impacts on improved behavior in the long-term effects. The major changes pertaining to the injury, such as inflammation and ROS production, start immediately after the injury and peak from within a few hours to 3 days after MCAO and other severe brain injuries [27]. Considering this, we chose the 6 h, 24 h, and 48 h timepoints to introduce the LIF and measure the mitochondrial function at 72 h. Still, in future, it is necessary to optimize the therapeutic window of intervention to have the maximum protection against injury. LIF-treated male rats showed significantly improved mitochondrial respiration in the ipsilateral striatum compared with PBS-treated rats, particularly in both coupled and uncoupled states of respiration (State III, IV, and V-CII). A noteworthy reversal of typical ischemic suppression was observed in State III, which was also an indicator of the ATP production in the striatum in LIF-treated males, compared with PBS-treated males, suggesting potentially enhanced energy efficiency. Ipsilateral prefrontal cortex regions of male rats showed slightly less significant injury effects than in the striatum, suggesting a more localized, subcortical ischemic injury. Likewise, male prefrontal cortex regions showed a less pronounced effect of LIF treatment on mitochondrial function. Numerical trends suggested some improvement in the prefrontal cortex, though none were found to be significant. This suggests that the protective effects of LIF in males may be stronger in subcortical regions like the striatum, or that this region specificity could be related to the pathophysiological progression of ischemia.

In female rats, the effects of LIF on mitochondrial respiration in both striatum and prefrontal cortex tissues were more variable and did not show improvement over PBS treatments. As compared to male rats, the female rats showed lower mitochondrial dysfunction when compared to their contralateral sides. These sex-specific differences could be attributed to the role of estradiol on the mitochondrial function in female rats [28]. While the female striatum also showed an injury effect in both treatment groups, OCR rates were not improved by LIF treatment compared to PBS treatment. In female ipsilateral prefrontal cortex regions, a significant injury effect was only observed in State V-CII in LIF-treated rats. This response in females could have been a result of sex-dependent neurotrophic factors modulating the efficacy of LIF treatment [15,29,30]. A primary mechanism of LIF’s protective effect is decreasing IFNγ expression, preventing the upregulation of CXCL10, an IFNγ-inducible chemokine [19]. Sex- and age-dependent differences in IFNγ activation has been well documented in humans and other mammals [31,32,33,34]. Previous studies have also demonstrated the efficacy of LIF treatment in reducing the cerebral edema volume in young male and aged female rat stroke models, though not in aged males [29]. This discrepancy may be attributed to age- associated decreases in LIFR expression in the brain, inhibiting LIF’s anti-oxidant enzyme production capacity. Sex-dependent LIF treatment responses may also result from hormonally regulated gene expression, potentially altering immune responses, LIFR upregulation, and cellular responses to cytokines. Collectively, these differences suggest that the neuroprotective effects of LIF are mediated by both age and sex.

LIF treatment has been previously reported to improve motor skill recovery in permanent large vessel stroke models in male rats [20]. This study also found that stroke-induced motor dysfunctions were attenuated by LIF-treatments in male, but not female, rats. The motor function assessments used in this study aimed to target striatum function in rats as this brain region is associated with motor coordination, and serves as a relevant model for better understanding stroke pathology and recovery mechanisms. Collectively, male rats treated with LIF showed numerically improved motor coordination compared to PBS-treated males in paw extension tests. However, PBS-treated females showed numerically improved motor coordination compared to LIF-treated females in both circling and paw extension tests. Overall, females exhibited a higher instance of typical healthy behaviors post MCAO compared to males with respect to circling behavior and paw extension capability. Future studies, including long-term investigations of tissue-specific and sex-specific secondary ischemic injury, along with other behavioral tests related to cognition and emotion, may provide insight into the chronic pathophysiology of stroke symptoms. Another important question concerns how LIF treatment is linked to mitochondrial function protection. We believe that LIF may be functional at two levels. First, as an anti-oxidant, it will directly protect mitochondria and help restore energy production. Second, it suppresses the inflammatory response of microglial activation, which also involves the bioenergetic switch. Additionally, we do not rule out the possibility that LIF may be targeting the signaling pathways other than mitochondrial bioenergetic function. Since we observed significant improvements in mitochondrial function with LIF treatment in the MCAO model, future studies will focus on how LIF affects bioenergetic function in different types of cells and how this translates into improved neuronal output.

## 5. Conclusions

In conclusion, this study highlighted the neuroprotective role of LIF in mitigating ischemia-induced mitochondrial dysfunction, particularly in male rats, where it significantly improved mitochondrial respiration in the striatum. The findings suggest that LIF’s effects are both tissue- and sex-specific, with males exhibiting greater benefits in subcortical regions while females showed a more variable response, potentially influenced by hormonal factors. Additionally, LIF treatment improved motor function recovery in males but not in females. These results underscore the importance of considering sex and tissue specificity in post-stroke interventions and suggest LIF as a potential therapeutic biomolecule for mitochondrial bioenergetic protection and functional recovery following ischemic injury.

## Figures and Tables

**Figure 1 biomolecules-15-00738-f001:**
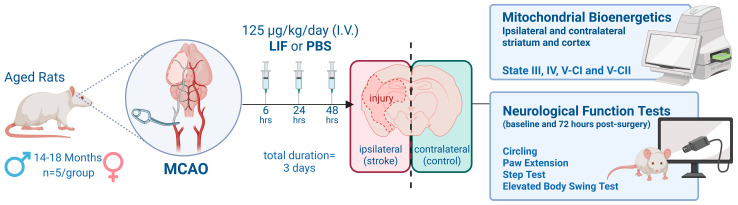
Conceptual diagram depicting study overview. Aged male and female rats (*n* = 19) underwent permanent middle cerebral artery occlusion (MCAO) procedures and were treated with LIF or PBS 6, 24, and 48 h after surgery. Animals were euthanized 72 h after surgery and brain tissues were collected. Mitochondrial bioenergetic parameters States III, IV, V-CI, and V-CII were measured in real time using a Seahorse XFe96 flux analyzer from isolated mitochondria in striatum and cortex regions of contralateral and ipsilateral hemispheres. Neurological scores were analyzed based on four behavioral tests (circling, paw extension, step test, and elevated body swing test) administered prior to surgery (baseline) and immediately before euthanasia (72 h post surgery).

**Figure 2 biomolecules-15-00738-f002:**
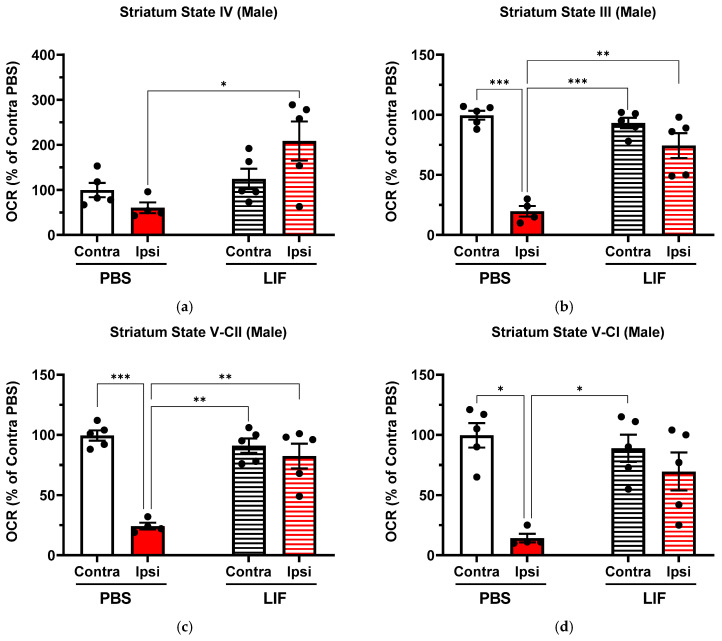
(**a**–**d**) Mitochondrial State III (**a**), State IV (**b**), State V-CI (**c**), and State V-CII (**d**) parameters in contralateral and ipsilateral striatum regions in male rats at 3 days post MCAO with solid columns representing PBS treatments and striped columns representing LIF treatments (*n* = 5 rats per treatment). * = *p* < 0.05, ** = *p* < 0.01, and *** = *p* < 0.001.

**Figure 3 biomolecules-15-00738-f003:**
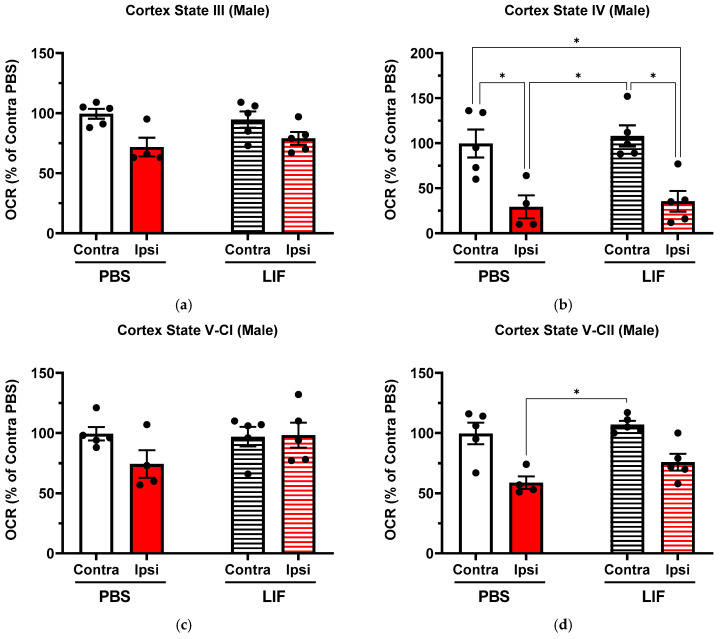
(**a**–**d**) Mitochondrial State III (**a**), State IV (**b**), State V-CI (**c**), and State V-CII (**d**) parameters in contralateral and ipsilateral cortex regions in male rats at 3 days post MCAO with solid columns representing PBS treatments and striped columns representing LIF treatments (*n* = 5 rats per treatment). * = *p* < 0.05.

**Figure 4 biomolecules-15-00738-f004:**
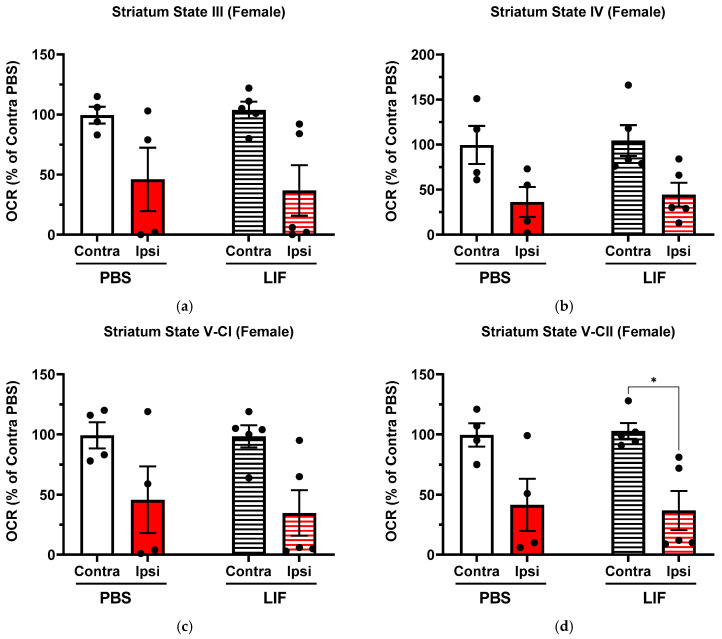
(**a**–**d**) Mitochondrial State III (**a**), State IV (**b**), State V-CI (**c**), and State V-CII (**d**) parameters in contralateral and ipsilateral striatum regions in female rats at 3 days post MCAO with solid columns representing PBS treatments and striped columns representing LIF treatments (*n* = 4–5 rats per treatment). * = *p* < 0.05.

**Figure 5 biomolecules-15-00738-f005:**
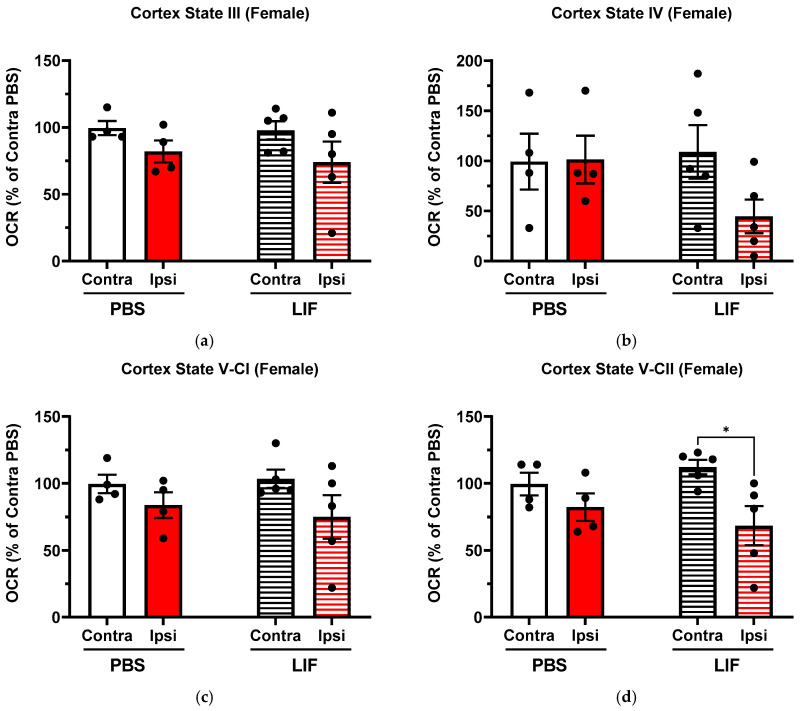
(**a**–**d**) Mitochondrial State III (**a**), State IV (**b**), State V-CI (**c**), and State V-CII (**d**) parameters in contralateral and ipsilateral cortex regions in female rats at 3 days post MCAO with solid columns representing PBS treatments and striped columns representing LIF treatments (*n* = 4–5 rats per treatment). * = *p* < 0.05.

**Figure 6 biomolecules-15-00738-f006:**
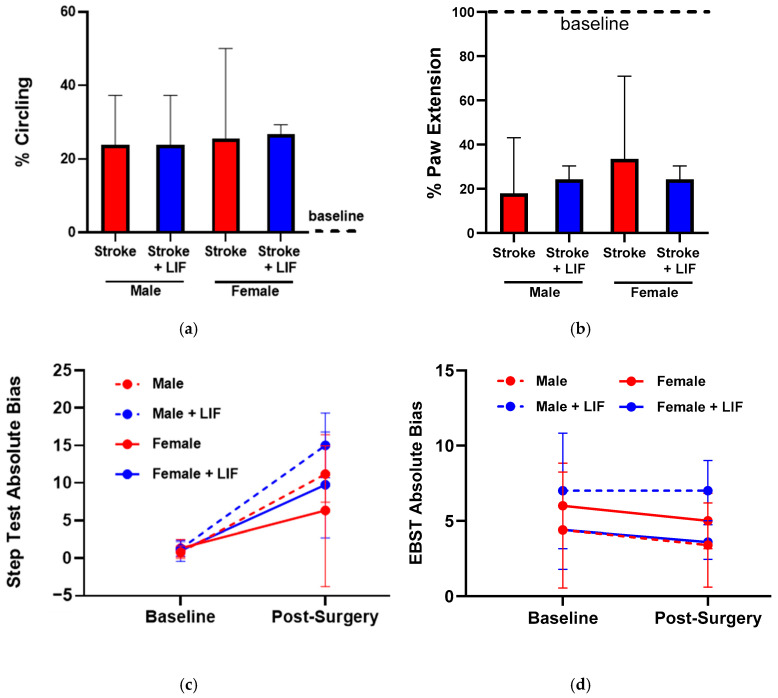
(**a**–**d**) Circling test results for LIF- and PBS-treated males and females (*n* = 4–5 rats per treatment) (**a**); ‘Yes’ responses indicate injury-induced circling behavior and ‘No’ responses indicate normal behavior. Paw extension test results for LIF- and PBS-treated males and females (*n* = 4–5 rats per treatment) (**b**). ‘Yes’ responses indicate normal ability to extend the paw and ‘No’ responses indicate an injury-induced inability to extend the paw. Step test absolute bias between left and right steps taken at baseline and post surgical testing (*n* = 4–5 rats per treatment) (**c**). Elevated body swing test absolute bias between left and right swings observed at baseline and post surgical testing (*n* = 4–5 rats per treatment) (**d**).

## Data Availability

Not applicable.

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
