# Peer review of "Sex- and Tissue-Specific Effects of Leukemia Inhibitory Factor on Mitochondrial Bioenergetics Following Ischemic Stroke"

_biomolecules, 2025, doi:10.3390/biom15050738_

Round 1
Reviewer 1 Report
Comments and Suggestions for Authors
The authors have presented solid research addressing the sex-specific role of LIF in improving ischemic damage and mitochondrial dysfunction in the MCAO rat model of stroke. This manuscript provides valuable insights and can be recommended for publication. However, I have outlined a few suggestions that I believe should be clarified or addressed:
- To simplify the presentation of results, I recommend combining the corresponding male and female graphs into a single panel, differentiating them by color (e.g., tones of blue for males, tones of pink for females).
- It seems that the authors could extract more information from the data obtained. For instance, presenting the ECAR parameter, the typical Seahorse curve, or an OCR/ECAR curve showing the four types of metabolic profiles could provide additional insights.
- The phrase, “…To model ischemic stroke in humans and to reflect the most critically affected human population…” is somewhat ambiguous. I recommend rephrasing it for clarity.
- “…To isolate mitochondria from rat brain tissue samples, the brain was quickly removed from euthanized animals and placed in a stainless-steel brain matrix…,” – Please specify the method of euthanasia used.
- « …LIF protects stroke induced decrease in mitochondrial respiration in striatum of male rats…. » - Probably, «Protects from…»
- I believe the authors could explain their rationale for selecting the prefrontal cortex (PFC) in their mitochondrial function studies. I noticed that, among the behavioral tests, there were no paradigms specific to the PFC. The prefrontal cortex is primarily associated with emotional behavior, learning and memory, as well as executive non-motor functions. To assess its function, tests such as the Y-maze, Plus-maze, prepulse inhibition, or NOR test could have been useful. This creates a discrepancy, as compared to the striatum, the PFC appears to be somewhat underexplored in this context.
Author Response
The authors have presented solid research addressing the sex-specific role of LIF in improving ischemic damage and mitochondrial dysfunction in the MCAO rat model of stroke. This manuscript provides valuable insights and can be recommended for publication. However, I have outlined a few suggestions that I believe should be clarified or addressed:
- Comment 1: To simplify the presentation of results, I recommend combining the corresponding male and female graphs into a single panel, differentiating them by color (e.g., tones of blue for males, tones of pink for females).
Response 1: Thank you for suggesting combining the male and female graphs. However, our representative path aligns with our hypothesis and experimental plan, where males were analyzed first, followed by females. Additionally, combining the graphs made it challenging to place significant comparison markers effectively.
- Comment 2: It seems that the authors could extract more information from the data obtained. For instance, presenting the ECAR parameter, the typical Seahorse curve, or an OCR/ECAR curve showing the four types of metabolic profiles could provide additional insights.
Response 2: We appreciate the reviewer’s comment for getting more information out of the experiments. In the present study, we have used the isolated mitochondria and not the intact cells. Due to lack of the cytosolic fraction, the ECAR data obtained in the assay are not relevant for the standard glycolytic assay as we do with the intact cells in the Seahorse.
- Comment 3: The phrase, “…To model ischemic stroke in humans and to reflect the most critically affected human population…” is somewhat ambiguous. I recommend rephrasing it for clarity.
Response 3: Thanks for the comment. We have rewritten the sentence for clarity.
[Change can be found – Page number: 3, Paragraph: 3, and Line: 121-124]
- Comment 4: “…To isolate mitochondria from rat brain tissue samples, the brain was quickly removed from euthanized animals and placed in a stainless-steel brain matrix…,” – Please specify the method of euthanasia used.
Response 4: Thanks for the comment. We have rewritten the sentence for clarity.
[Change can be found – Page number: 4, Paragraph: 5, and Line: 178-180]
- Comment 5: « …LIF protects stroke induced decrease in mitochondrial respiration in striatum of male rats…. » - Probably, «Protects from…»
Response 5: Thanks for the drawing our attention into this. We have corrected this to “LIF restores stroke induced decrease in mitochondrial respiration in striatum of male rats”
[Change can be found – Page number: 6, Paragraph: 1, and Line: 265]
- Comment 6: I believe the authors could explain their rationale for selecting the prefrontal cortex (PFC) in their mitochondrial function studies. I noticed that, among the behavioral tests, there were no paradigms specific to the PFC. The prefrontal cortex is primarily associated with emotional behavior, learning and memory, as well as executive non-motor functions. To assess its function, tests such as the Y-maze, Plus-maze, prepulse inhibition, or NOR test could have been useful. This creates a discrepancy, as compared to the striatum, the PFC appears to be somewhat underexplored in this context.
Response 6: We appreciate the reviewer’s comment regarding the importance of cognitive and emotional behavioral tasks in assessing prefrontal cortex-related functions. However, the circling behavior and paw extension tests are well-established methods for evaluating striatal and motor cortex impairments, which are key regions affected in our stroke model along with the prefrontal cortex. As this is a preliminary study focused on the association between LIF and mitochondrial function, our current approach prioritizes these standard motor assessments. In future studies, we plan to conduct a comprehensive therapeutic window analysis incorporating a broader range of behavioral assessments at both acute and chronic time points.
Author Note: While addressing the reviewers' comments, we added a few new references using PMID numbers. However, these have not yet been formatted according to the Biomolecules reference style, due to a broken EndNote link when preparing the manuscript in the Biomolecules Word format. We will ensure all references are properly formatted during the final proofreading stage. Additionally, reviewer responses have been highlighted in the draft for the convenience of the reviewers and editors, and will be cleaned up in the final version. Thank you.

Reviewer 2 Report
Comments and Suggestions for Authors
This study investigates the role of leukemia inhibitory factor in modulating mitochondrial bioenergetics in aged male and female rats following ischemic stroke. The use of a well-established middle cerebral artery occlusion model and mitochondrial respiration assessments provides valuable insights into the sex- and region-specific effects of LIF. The findings suggest that LIF improves mitochondrial function in the striatum of male rats but has less pronounced effects in females, contributing to the growing body of evidence on mitochondrial-targeted therapies for stroke. However, the study requires additional clarifications regarding the statistical power of sex-based subgroup analyses, particularly given the small sample sizes. Expanding on how these findings translate to potential clinical applications would improve the manuscript’s impact.
The methodology is detailed and appropriate, but certain aspects need refinement. The study successfully measures mitochondrial respiration using the Seahorse Flux Analyzer, yet does not sufficiently discuss the potential confounding effects of ischemia-induced metabolic alterations on oxygen consumption rates. Additionally, the rationale behind selecting specific time points for LIF administration and mitochondrial assessments should be further justified. The behavioral testing outcomes, though informative, are not comprehensively linked to the mitochondrial findings, making it unclear whether the observed bioenergetic improvements translate into functional recovery. A more robust discussion on how LIF mechanistically interacts with mitochondrial signaling pathways in ischemic conditions would strengthen the manuscript’s depth.
The discussion appropriately compares the findings to previous studies on LIF and neuroprotection, but it would benefit from a more critical evaluation of potential limitations. The sex-based differences in LIF efficacy warrant further exploration, particularly considering hormonal influences on mitochondrial function. Additionally, the study does not thoroughly address how LIF’s systemic effects might contribute to its neuroprotective action beyond direct mitochondrial modulation. Future studies incorporating additional time points, mechanistic validations, and a broader range of outcome measures would help reinforce the study’s conclusions.
Author Response
Comments: This study investigates the role of leukemia inhibitory factor in modulating mitochondrial bioenergetics in aged male and female rats following ischemic stroke. The use of a well-established middle cerebral artery occlusion model and mitochondrial respiration assessments provides valuable insights into the sex- and region-specific effects of LIF. The findings suggest that LIF improves mitochondrial function in the striatum of male rats but has less pronounced effects in females, contributing to the growing body of evidence on mitochondrial-targeted therapies for stroke. However, the study requires additional clarifications regarding the statistical power of sex-based subgroup analyses, particularly given the small sample sizes. Expanding on how these findings translate to potential clinical applications would improve the manuscript’s impact.
The methodology is detailed and appropriate, but certain aspects need refinement. The study successfully measures mitochondrial respiration using the Seahorse Flux Analyzer, yet does not sufficiently discuss the potential confounding effects of ischemia-induced metabolic alterations on oxygen consumption rates. Additionally, the rationale behind selecting specific time points for LIF administration and mitochondrial assessments should be further justified. The behavioral testing outcomes, though informative, are not comprehensively linked to the mitochondrial findings, making it unclear whether the observed bioenergetic improvements translate into functional recovery. A more robust discussion on how LIF mechanistically interacts with mitochondrial signaling pathways in ischemic conditions would strengthen the manuscript’s depth.
The discussion appropriately compares the findings to previous studies on LIF and neuroprotection, but it would benefit from a more critical evaluation of potential limitations. The sex-based differences in LIF efficacy warrant further exploration, particularly considering hormonal influences on mitochondrial function. Additionally, the study does not thoroughly address how LIF’s systemic effects might contribute to its neuroprotective action beyond direct mitochondrial modulation. Future studies incorporating additional time points, mechanistic validations, and a broader range of outcome measures would help reinforce the study’s conclusions.
Responses: Thank you very much for critically evaluating the manuscript. We have thoroughly reviewed the comments and made the following changes to the manuscript.
- We have included the power analysis statement in the statistics analysis.
[Change can be found – Page number: 3, Paragraph: 3, and Line: 121-125]
- We have included in the introduction about the role of mitochondrial dysfunction in driving the neuronal damage in brain injury.
[Change can be found – Page number: 2, Paragraph: 2, and Line: 65-74]
- We have updated the discussion with the rationale for choosing the specific timepoints for the LIF therapy (Lines 416-420).
[Change can be found – Page number: 13, Paragraph: 1, and Line: 412-421 and 424-426]
- We have included the limitations and future prospects of the study to address the related concerns on possible off targets of LIF treatment (Lines 466-477)
[Change can be found – Page number: 14, Paragraph: 1, and Line: 466-477]
Author Note: While addressing the reviewers' comments, we added a few new references using PMID numbers. However, these have not yet been formatted according to the Biomolecules reference style, due to a broken EndNote link when preparing the manuscript in the Biomolecules Word format. We will ensure all references are properly formatted during the final proofreading stage. Additionally, reviewer responses have been highlighted in the draft for the convenience of the reviewers and editors, and will be cleaned up in the final version. Thank you.
Round 2
Reviewer 2 Report
Comments and Suggestions for Authors
the authors improved the paper.
Author Response
All necessary changes have been made to the final draft, including improving the quality of figures and adding references.
